# Association of Night Eating Habits with Health-Related Quality of Life (HRQoL) in University Students

**DOI:** 10.3390/healthcare10040640

**Published:** 2022-03-28

**Authors:** Yoonji Kim, Jung Hyun Kwak, Jean Kyung Paik

**Affiliations:** Department of Food and Nutrition, Eulji University, Seongnam 13135, Korea; kyoonj@citizen.seoul.kr (Y.K.); hyun4615@hanmail.net (J.H.K.)

**Keywords:** students, health-related quality of life, eating habits, night eating

## Abstract

University students have dietary habits such as irregular eating habits, night eating habits (NEHs), and alcohol drinking, which can lead to nutritional problems. Especially, NEHs may increase the risk of chronic diseases and reduce the health-related quality of life (HRQoL). The purpose of this study was to investigate associations between NEHs and HRQoL and to evaluate if their associations may differ according to gender. Data were obtained from Eulji University March-April 2018. Participants were recruited via advertisements in school. The questionnaire, including socio-demographic factors, dietary and NEHs, and HRQoL was self-administered by participants. We found that, in males, when subjects with night eating frequency ≥1 time/week, the HRQoL showed a tendency to deteriorate compared to those with night eating frequency <1 time/week. In females, when the subject answered that they consumed a lot of night meal intake, their HRQoL was significantly deteriorated compared to those who answered that they consumed a moderate or small. Therefore, we confirmed that there is a slight difference in the factors of NEHs that affect the HRQoL by gender.

## 1. Introduction

Night eating represents a necessary behavioral symptom of night eating syndrome (NES), a disorder characterized by evening hyperphagia, morning anorexia, and sleep disturbance [1]. According to previous studies, night eating habits (NEHs) were associated with a greater risk of chronic diseases such as obesity [2], metabolic syndrome [3], and digestive system disease [4]. In particular, university students with NEHs had poorer eating habits than those without NEHs because of frequently skipping breakfast, alcohol consumption, and high fat intake [5]. A study reported that younger adults (age 18–30) had higher prevalence of night eating than any other age group [6]. University students are at the age of transition from adolescence to adulthood, and poor eating behavior at this time can lead to health problems in adulthood [5]. Thus, practicing healthy eating habits during university is crucial to successful aging.

University students are exposed to a lot of stress due to problems such as school schedule and employment and have unhealthy dietary habits such as irregular eating habits, NEHs, and alcohol drinking [7]. NEHs may increase the risk of chronic diseases such as hypertension, dyslipidemia, stroke, and myocardial infarction, which could reduce the health-related quality of life (HRQoL) [8]. HRQoL reflects the individual’s perception of comprehensive general health status, including physical, mental, and social functions [9].

According to previous studies [10,11], females have worse HRQoL than males, and possible reasons include differences in socioeconomic and physiologic factors [12]. However, studies on the association of NEHs on HRQoL by gender are insufficient. We hypothesized that the HRQoL might deteriorate in university students with NEHs, and the associations may differ by gender.

Thus, this study investigated the associations between NEHs and HRQoL and evaluated if their associations may differ by gender among university students.

## 2. Materials and Methods

### 2.1. Subjects

Data were obtained from Eulji University March–April 2018. The participated in our study were students from a private university in Gyeonggi-do, Korea. The total number of students was 4731 (as of 2021), and majors were diverse, including humanities, social studies, natural sciences, and engineering. Participants were convenience samples that were easily recruited from within the university using advertisements in school. Inclusion criteria were: (1) those who have little or no restrictions on their eating habits due to other intentions; (2) who can choose and cook their own food; and (3) who are 19–29 years old. A total of 129 subjects were recruited, and 126 subjects were included in the analysis, excluding 3 subjects who had insufficient questionnaire responses.

Prior to participation in this study, all participants provided written informed consent. The Eulji University Institutional Review Board (reference number EU18-20) approved this study. The questionnaire, including socio-demographic factors such as gender (male and female), age (19–29), grade (1–4), height, weight, type of residence (home, others), pocket money (per month) (less than USD 83.8; KRW 100,000, USD 83.8–167.6; KRW 100,000–200,000, USD 167.6–251.4; KRW 200,000–300,000, USD 251.4–335.2; KRW 300,000–400,000, more than USD 335.2; KRW 400,000), disease status (digestive organ, infectious, nervous, liver, fatigue, other, none), and wake up time (before 7 a.m., 7–8 a.m., after 8 a.m.) was self-administered by participants. Body mass index (BMI, kg/m^2^) at diagnosis was calculated by dividing participants’ weight (kg) by their height (m^2^).

### 2.2. Assessment of Night Eating Habits and Dietary Factor

A dietary-related questionnaire [13,14] including the frequency of meals (breakfast, lunch, dinner), if they ate with others, quantity of meals, times of meals, picky eating, preferred taste, and frequency of alcohol intake was self-administered by participants.

Questions 1–14 [15,16] queried NEHs using multiple choice, including the following subjects: (1) frequency of night eating, (2) reason for night eating, (3) quantity of night meals, (4) factors influencing night eating, (5) night eating with companion(s), (6) accompanying drinks when night eating, (7) drink-related questions, (8) preferences by type of night eating, (9) preferred taste, (10) considerations in selecting night eating, (11) place of night eating, (12) night eating time, (13) endpoint of night eating, (14) time went to bed after night eating (Appendix A).

### 2.3. Assessment of Health-Related Quality of Life

HRQoL was measured using an 8-item HRQoL Instrument (HINT-8) suggested by the Korea Centers for Disease Control and Prevention (KCDC) [17]. The HINT-8 measures consist of eight questions (climbing stairs, pain, energy, working, depression, memory, sleeping, and happiness) focused on the self-perception of general health, physical health, mental health, and the limitations on activity. Each question was addressed on a four-point Likert scale with a higher score indicating worse satisfaction with their HRQoL. The selected scoring method was standard (the responses were transformed into ranks: a to 1 (lowest agreement), b to 2, c to 3, and d to 4 (highest agreement)). The maximal total scores of HRQoL questionnaires were 32.

### 2.4. Statistical Analyses

The Statistical Package for the Social Sciences (SPSS) 20.0 (SPSS Inc., Chicago, IL, USA) statistical software package was used for all statistical calculations. Data are presented as the mean ± standard deviation (SD) for continuous variables. Independent *t*-tests were used to compare continuous variables by gender. The chi-square test was used to analyze the differences in NEHs by gender. We confirmed that the HRQoL followed a normal distribution. The relationship between NEHs and HRQoL measurement was analyzed using multiple regression analysis. The variables related to the NEHs were generated and analyzed as dummy variables. The values of *p* < 0.05 (2-tailed) were statistically significant.

## 3. Results

### 3.1. Descriptive Statistics

Table 1 shows the general characteristics by gender. In males, the mean (±SD) age was 23.3 (±2.2), and the means (±SD) of height and weight were 175.8 (±4.9) cm and 72.4 (±11.8) kg, respectively. In females, the mean (±SD) age was 21.7 (±1.1), and the means (±SD) of height, and weight were 161.5 (±5.0) cm and 54.0 (±6.8) kg, respectively. Males were significantly older and had significantly higher weight than females. Additionally, BMI, pocket money, average meal type, and picky eating were significantly different depending on gender.

Table 2 shows the distribution of NEHs by gender. The reasons for night eating and time went to bed after eating were significantly different depending on gender. A high proportion of males responded that the reason for night eating was because they were hungry. A high proportion of females responded that they went to bed more than two hours after having a night eating.

### 3.2. Night Eating Habits and HRQoL Group by Gender

Table 3 presents the frequency of the HRQoL group and NEHs by gender. There was a significant difference in the distribution of HRQoL with night eating frequency and whether or not to eat with others in males.

### 3.3. Night Eating Frequency and HRQoL by Gender

Table 4 shows the results of the linear regression analyses for the relationship of night eating frequency with HRQoL by gender. In males, subjects with night eating frequency ≥ 1 time/week had a tendency higher HRQoL 1.49 (±0.84) than those with night eating frequency <1 time/week. No statistical significance was observed in night eating frequency and HRQoL in females. 

### 3.4. Night Meal Intake and HRQoL by Gender

Table 5 shows the results of the linear regression analyses for the association of night meal intake with HRQoL stratified by gender. Females who answered that they had a high intake of night meals had a significantly higher HRQoL 2.88 (±1.13) than those who answered that they had a lower intake of night meals. No statistical significance was observed in night meal intake and HRQoL in males.

## 4. Discussion

In our study, we found that there is a slight difference in the factors of NEHs that affect the HRQoL by gender. The HRQoL deteriorated when the night meal intake was high in females. 

Several studies reported that younger adults had problems in dietary-related habits such as irregular and skipped meals, picky eating, fast food intake, and low nutrient density [18,19]. College students are at an age where they are transitioning from adolescence to adulthood, physically and socially. Poor eating behavior at this time can lead to health problems [5]. Lim et al. investigated the bone mineral density (BMD) of college students to analyze the differences in BMD according to lifestyle. They found college students who often ate fast-food had significantly lower BMD than those who did not eat it at all [20]. Additionally, Kim et al. reported that female college students with binge eating disorder (BED) had higher levels of functional impairment and mental health problems [21]. Particularly, NEHs were associated with a greater risk of chronic diseases such as obesity [2], metabolic syndrome [3], and digestive system diseases [4]. In a longitudinal study, Yoshida et al. reported that subjects with NEHs had higher odds for obesity and were associated with dyslipidemia [22]. NEHs increase the risk of chronic diseases, which could reduce the HRQoL [8]. 

According to the World Health Organization (WHO), quality of life is defined as an “individuals’ position in life in the context of the culture and value systems in which they live and in relation to their goals, expectations, standards and concerns” [23]. Various factors such as physical activity, sedentary time, and sleep duration can influence the HRQoL [24,25,26]. A recent study by Ge et al. suggested that increasing physical activity and promoting adequate sleep duration may have a positive impact on the HRQoL of college students [27]. Additionally, Kim et al. observed that low sedentary behavior and moderate-to-high vigorous physical activity were associated with a decreased risk of poor HRQoL in a representative sample of U.S. adults [28]. However, the effects of dietary factors such as NEHs on health-related quality of life have not been reported in healthy Korean. College students are exposed to a lot of stress due to problems such as school schedule and employment and have dietary habits such as irregular eating habits, NEHs, and alcohol drinking [7]. Thus, to improve the HRQoL in university students, it is necessary to evaluate which dietary factors affect the HRQoL. According to previous studies [10,11], females have worse HRQoL than males, and possible reasons include differences in socioeconomic and physiologic factors [29]. In our study, we found a significant difference in the comparison of the average HRQoL between males and females. Additionally, in females, a high intake of night eating deteriorated the HRQoL.

In a study by Seo et al. conducted to investigate the dietary behaviors and stress-related factors among college students, they found problems such as overeating in females as eating behavior to relieve stress [30]. Even in the same age group, there were differences between males and females in how they deal with stress. Similar to previous research, the proportion of females who responded to eating night meals to relieve stress was higher than males in our study. Therefore, university students with NEHs could affect their quality of life; it is necessary to provide guidelines to reduce the frequency and amount of night eating by gender.

Additionally, in our study, we did not confirm whether NEHs directly affect chronic diseases such as hypertension, diabetes, and hypercholesterolemia. Since the subject of our study is university students (average age: 23.3 years for males, 21.7 years for females) the prevalence of chronic diseases is not expected to be high. According to Korea National Health Statistics (2020), the prevalence of hypertension, diabetes, and hypercholesterolemia among 19–29-year-olds was 7.4%, 0.7%, and 5.2%, respectively, and the prevalence of these chronic diseases increased after the age of 40. Therefore, continuous nutrition education and efforts to improve eating habits will be required in university students because, if an unhealthy dietary habit such as NEHs is continued, it could increase chronic diseases after middle age and act as a factor that lowers the HRQoL. 

The limitations of our study are as follows. (1) This study used a questionnaire to evaluate the night eating habits of university students. (2) The sizes of the samples were small, and because of the study’s cross-sectional design, we cannot demonstrate causality. (3) There is still a possibility of residual confounding. (4) The recruitment type of our study has limitations because participants were a convenience sample that was recruited within the university.

Nevertheless, this study can provide information on dietary habits in university students and is the first to our knowledge that demonstrates the association between NEHs and HRQoL by gender.

## 5. Conclusions

In conclusion, we found that there is a slight difference in the factors of NEHs that affect the HRQoL by gender. Therefore, we suggested it may help to reduce the intake of night meals in females to improve the HRQoL. In addition, it is thought that it is necessary to improve the quality of life through proper nutrition education considering the NEHs characteristics by gender so that college students do not relieve their stress by NEHs. However, further studies are needed to determine the clinical significance of night eating habits in sufficient samples.

## Figures and Tables

**Table 1 healthcare-10-00640-t001:** General characteristics of participants.

	Males (n = 52)	Females (n = 74)	*p*
Age (years)	23.3 ±2.2	21.7 ± 1.1	<0.001
Height (cm)	175.8 ± 4.9	161.5 ± 5.0	0.854
Weight (kg)	72.4 ± 11.8	54.0 ± 6.8	0.001
HRQOL score	12.1 ± 0.42	14.5 ± 0.40	<0.001
BMI (kg/m^2^)	23.4 ± 3.2	20.7 ± 2.3	<0.001
BMI category			
Underweight (<18.5)	1 (1.9)	11 (15.1)	<0.001
Normal (18.5–22.9)	7 (51.9)	49 (67.1)	
Overweight (23–24.9)	13 (25.0)	10 (13.7)	
Obesity (≥25)	11 (21.2)	3 (4.1)	
Residential type			
Home	37 (71.2)	58 (78.4)	0.404
Others	15 (28.8)	16 (21.6)	
Pocket money (per month)			
<$83.8	18 (34.6)	13 (17.6)	0.003
$83.8–167.6	7 (13.5)	9 (12.2)	
$167.6–251.4	12 (23.1)	10 (13.5)	
$251.4–335.2	9 (17.3)	22 (29.7)	
>$335.2	6 (11.5)	20 (27.0)	
Wake up time			
Before 7 a.m.	18 (34.6)	21 (28.4)	0.924
7–8 a.m.	16 (30.8)	33 (44.6)	
After 8 a.m.	18 (34.6)	20 (27.0)	
Average meal time			
Less than 10 min	11 (21.2)	5 (6.8)	0.019
10–20 min	34 (65.4)	52 (70.3)	
More than 20 min	7 (13.5)	17 (23.0)	
Picky eating			
No	18 (34.6)	6 (8.1)	0.033
Little Not	12 (23.1)	30 (40.5)	
No opinion	15 (28.8)	27 (36.5)	
Yes	7 (13.5)	11 (14.9)	
Frequency of drinking (per week)			
Almost do not	7 (13.5)	10 (13.5)	0.099
1–2 times	20 (38.5)	21 (28.4)	
3–4 times	20 (38.5)	22 (29.7)	
More than 5–6 times	5 (9.6)	21 (28.4)	

Independent *t*-test for continuous variable and Chi-square test for category variable were used. *p* < 0.05 considered significant.

**Table 2 healthcare-10-00640-t002:** Distribution of night eating habits by gender.

	Males (n = 52)	Females (n = 73) *	*p*
Night eating frequency			
<1 time/week	26 (50.0)	26 (35.6)	0.108
≥1 time/week	26 (50.0)	47 (64.4)	
Night meal intake			
Small and Moderate	41 (78.8)	63 (86.3)	0.272
Large	11 (21.2)	10 (13.7)	
With others			
Alone	16 (30.8)	19 (26.0)	0.561
Others	36 (69.2)	54 (74.0)	
Night eating time			
<10 p.m.	9 (17.3)	19 (26.0)	0.296
10 p.m.–12 midnight	37 (71.2)	47 (64.4)	
>12 midnight	6 (11.5)	7 (9.6)	
Reasons for night eating *			
Hungry	29 (76.3)	36 (56.3)	0.042
Others	9 (23.7)	28 (43.8)	
Time went to bed after eating			
<1 h	17 (32.7)	6 (8.2)	<0.001
1–2 h	26 (50.0)	33 (45.2)	
>2 h	9 (17.3)	34 (46.6)	

Chi-square test for category variable were used. *p* < 0.05 considered significant. * Those who did not respond were excluded.

**Table 3 healthcare-10-00640-t003:** Distribution of night eating habits and HRQoL group by gender.

	Males (n = 52)			Females (n = 73) *		
	HRQOL <Average Score	HRQOL ≥Average Score	*p*	HRQOL <Average Score	HRQOL ≥Average Score	*p*
Night eating frequency						
<1 time/week	16 (61.5)	10 (34.5)	0.012	16 (41.0)	10 (29.4)	0.301
≥1 time/week	7 (30.4)	19 (65.5)		23 (59.0)	24 (70.6)	
Night meal intake						
Small and Moderate	19 (82.6)	22 (75.9)	0.554	36 (92.3)	27 (79.4)	0.110
Large	4 (17.4)	7 (24.1)		3 (7.7)	7 (20.6)	
With others						
Alone	2 (8.7)	14 (48.3)	0.002	10 (25.6)	9 (26.5)	0.936
Others	21 (91.3)	15 (51.7)		29 (74.4)	25 (73.5)	
Night eating time						
<10 p.m.	5 (21.7)	4 (13.8)	0.386	10 (25.6)	9 (26.5)	0.811
10 p.m.–12 midnight	16 (69.6)	21 (72.4)		26 (66.7)	21 (61.8)	
≥12 midnight	2 (8.7)	4 (13.8)		3 (7.7)	4 (11.8)	
Reasons for night eating *						
Hungry	10 (71.4)	19 (79.2)	0.588	19 (54.3)	17 (58.6)	0.728
Others	4 (28.6)	5 (20.8)		16 (45.7)	12 (41.4)	
Time went to bed after eating						
<1 h	7 (30.4)	10 (34.5)	0.829	4 (10.3)	2 (5.9)	0.702
1–2 h	12 (52.2)	14 (48.3)		15 (38.5)	18 (52.9)	
≥2 h	4 (17.4)	5 (17.2)		20 (51.3)	14 (41.2)	

Chi-square test for category variable were used. *p* < 0.05 considered significant. * Those who did not respond were excluded.

**Table 4 healthcare-10-00640-t004:** Adjusted linear regression and 95% confidence interval (95%CI) of HRQoL by night eating frequency.

	HRQoL			
	Model A		Model B	
	β ± S.E	*p*	β ± S.E	*p*
Males				
<1 time/week	0 (ref)	-	0 (ref)	-
≥1 time/week	1.42 ± 0.61	0.023	1.49 ± 0.84	0.080
Females				
<1 time/week	0 (ref)	-	0 (ref)	-
≥1 time/week	0.80 ± 0.84	0.344	0.74 ± 0.86	0.396

*p* < 0.05 considered significant. Linear regression for continuous variable was used. Model A was nonadjusted model. Model B: model A + further adjusted for age, BMI.

**Table 5 healthcare-10-00640-t005:** Adjusted linear regression and 95% confidence interval (95%CI) of HRQoL by night meal intake.

	HRQoL			
	Model A		Model B	
	β ± S.E	*p*	β ± S.E	*p*
Males				
Small and Moderate	0 (ref)	-	0 (ref)	-
Large	0.52 ± 1.03	0.615	0.26 ± 1.07	0.811
Females				
Small and Moderate	0 (ref)	-	0 (ref)	-
Large	2.76 ± 1.14	0.018	2.88 ± 1.13	0.013

*p* < 0.05 considered significant. Linear regression for continuous variable was used. Model A was nonadjusted model. Model B: model A + further adjusted for age, BMI.

## Data Availability

The data presented in this study are available upon request from the corresponding authors. The data are not publicly available because of the privacy concerns of the subjects.

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
