# Peer review of "Association of Night Eating Habits with Health-Related Quality of Life (HRQoL) in University Students"

_healthcare, 2022, doi:10.3390/healthcare10040640_

Round 1

Reviewer 1 Report

The manuscript was well improved. However, some changes are needed before its approval. 

  • Line 53 - 55: Describe the University's characteristics. How many students? What kind of courses?
  • How did you recruit the participants? 
  • The inclusion criteria should be mentioned prior to the number of participants
  • Insert the conversion of pocket money also in your currency.
  • Lines 74 - 79: Include the questionnaire as a supplementary file.
  • Lines 199 - 202: include the limitation on recruitment type.

Thank you for the opportunity to review this manuscript!

Author Response

We sincerely appreciate the time spent in reviewing this manuscript and your advice to improve it. We revised and supplemented the reviewer's comments in the text.

Reviewer1

The manuscript was well improved. However, some changes are needed before its approval. 

  • Line 53 - 55: Describe the University's characteristics. How many students? What kind of courses?

:Total number of students is 4731 (as of 2021), and majors are diverse, including humanities, social studies, natural sciences, and engineering.

  • How did you recruit the participants? 

: Participants were recruited via advertisements in school.

  • The inclusion criteria should be mentioned prior to the number of participants

: We revised the position of the sentence.  

  • Insert the conversion of pocket money also in your currency.

: We have added our currency units as follows:

“(less than $83.8; 100,000 won, $83.8-167.6; 100,000-200,000 won, $167.6-251.4; 200,000-300,000 won, $251.4-335.2; 300,000-400,000 won, more than $335.2; 400,000 won)”

  • Lines 74 - 79: Include the questionnaire as a supplementary file.

: We have added the questionnaire. However, since this is a Korean version of the questionnaire, if you need an English version, please let us know.

  • Lines 199 - 202: include the limitation on recruitment type.

:We have added the recruitment type as a limitation as follows:

“The recruitment type of our study has limitation because participants were a convenience sample that was easily recruited within the university”

Reviewer 2 Report

The Authors included my suggestions in the manuscript, so I think that the publication is suitable for printing.

Author Response

We sincerely appreciate the time spent in reviewing this manuscript and your advice to improve it